# Catalogue for Transmission Genetics in Arabs (CTGA) Database: Analysing Lebanese Data on Genetic Disorders

**DOI:** 10.3390/genes12101518

**Published:** 2021-09-27

**Authors:** Sami Bizzari, Pratibha Nair, Asha Deepthi, Sayeeda Hana, Mahmoud Taleb Al-Ali, André Megarbané, Stephany El-Hayek

**Affiliations:** 1Centre for Arab Genomic Studies, Dubai 22252, United Arab Emirates; sami@hmaward.org.ae (S.B.); pratibha.nair@gmail.com (P.N.); ashadeepthi22@gmail.com (A.D.); sayeeda.hana@gmail.com (S.H.); mtalali@gmail.com (M.T.A.-A.); 2Department of Human Genetics, Gilbert and Rose-Marie Chagoury School of Medicine, Lebanese American University, Beirut 13-5053, Lebanon; andre.megarbane@lau.edu.lb

**Keywords:** Lebanon, CTGA, database, gene, genetic disorder, variants, rare diseases, consanguinity

## Abstract

Lebanon has a high annual incidence of birth defects at 63 per 1000 live births, most of which are due to genetic factors. The Catalogue for Transmission Genetics in Arabs (CTGA) database, currently holds data on 642 genetic diseases and 676 related genes, described in Lebanese subjects. A subset of disorders (14/642) has exclusively been described in the Lebanese population, while 24 have only been reported in CTGA and not on OMIM. An analysis of all disorders highlights a preponderance of congenital malformations, deformations and chromosomal abnormalities and demonstrates that 65% of reported disorders follow an autosomal recessive inheritance pattern. In addition, our analysis reveals that at least 58 known genetic disorders were first mapped in Lebanese families. CTGA also hosts 1316 variant records described in Lebanese subjects, 150 of which were not reported on ClinVar or dbSNP. Most variants involved substitutions, followed by deletions, duplications, as well as in-del and insertion variants. This review of genetic data from the CTGA database highlights the need for screening programs, and is, to the best of our knowledge, the most comprehensive report on the status of genetic disorders in Lebanon to date.

## 1. Introduction

Lebanon is a Levantine country, located on the Mediterranean Sea. Although a small country, with a total area of 10,452 km^2^, its location on the crossroad of both land and maritime routes has provided it with great historical and cultural significance since antiquity. The total population in Lebanon is currently estimated at 6.8 million people [1]. However, non-Lebanese residents account for almost 30% of this figure, with the massive influx of refugees, mostly from Syria, that the country has witnessed in the past decade [2]. On the other hand, during and following the Lebanese civil war and up until the present, a significant proportion of citizens left the country, contributing to a large Lebanese diaspora (estimated 4–13 million individuals) across the globe, especially concentrated in South America, Canada, and Australia [3].

Genetic disorders are highly prevalent in the Arab World [4,5,6]. Some of the factors contributing to this high prevalence in these populations include the high fertility and birth rates, and the high rates of consanguineous unions [7]. Although lower relative to countries in the Arabian Gulf and North Africa, Lebanon still has a high annual incidence of birth defects at 63 per 1000 live births; most of these congenital defects being due to genetic factors [4]. Genetic disorders in Lebanon have been overviewed by Nakouzi et al., Khneisser et al., and earlier by Der Kaloustian [8,9,10].

The Catalogue for Transmission Genetics in Arabs (CTGA) database, hosted online at https://cags.org.ae/ (accessed on 29 July 2021), is a compendium of bibliographic data on genetic disorders in Arabs, compiled and curated through searches on PubMed and Index Medicus [11]. To date, CTGA is the only database documenting Arab genomic variation which is entirely manually curated. In this review, we present a report of genetic disorders and associated gene variants in the Lebanese population pulled from the CTGA database. This is, to the best of our knowledge, the most comprehensive review on the status of genetic disorders in this country.

## 2. Materials and Methods

We conducted a comprehensive literature search on PubMed for biomedical literature either originating from Lebanon or referring to Lebanese subjects, using the search string “Lebanon* OR Lebanese*”, up to the end of 2020. The 22,159 articles thus obtained were manually screened according to the following inclusion criteria: (a) article describing a genetic disease in a Lebanese individual(s) or subject(s) of Lebanese origin, (b) article describing gene variant(s) in a Lebanese individual or in the Lebanese population, and (c) article reporting a disorder, not commonly known to be genetic, in multiple members of a Lebanese family. The following categories of articles were excluded: (a) articles carrying information only on non-Lebanese subjects, (b) articles carrying redundant data. The 814 screened articles were then carefully analyzed and information on the genetic disorder, relevant HPO terms, and gene variants reported in anonymous Lebanese subjects were manually extracted and added to the CTGA database (Figure 1A).

CTGA is an SQL database containing five related categories of records: disease, gene, variant, subject, and reference article. Disease and gene records are linked to their corresponding OMIM records, whenever available. Published variants for which HGVS reference sequence positions were identifiable were entered into CTGA as distinct records and linked to relevant subjects. These variants are linked to their dbSNP and ClinVar records, if available. Data on other variants were incorporated in text descriptions in relevant gene and disease records, but not as separate variant records. Anonymous subject records contain HPO terms and are linked to the relevant published article reference. Data can be accessed through a simple or advanced search.

## 3. Results and Discussion

Although genetic publications on Lebanese subjects started as early as 1950 [12], it is only after the late 1990s, following the end of the Lebanese civil war, that we see a significant amount of genetic literature being published. An earlier study analyzing the biomedical bibliometric output from Lebanon until 2007 showed an increasing trend for publications [13]. We see a similar trend in publications over the years in our analysis of year-wise distribution of the selected articles, signifying an increase in genetic research. We were also interested in monitoring the evolution of medical genetic studies, especially with the advent of newer molecular technologies. The percentage of research articles with available molecular data shows a clear increasing trend, especially over the last decade (Figure 1B), corresponding to the time period when NGS techniques were adopted by clinicians in Lebanon for the diagnosis of genetic diseases. Interestingly, the number of such publications declined in the past couple of years, likely due to the economic crisis that Lebanon has since been facing.

The CTGA database currently holds data on 642 genetic diseases, 676 related genes, and 1316 variants described in the Lebanese population. Of all diseases, 24 are genetic and/or congenital disorders that are not available on OMIM (Table 1). Almost all of these are syndromic conditions, with a combination of clinical features not described elsewhere. For instance, variants of MCA/MR (multiple congenital anomalies/mental retardation) conditions with new constellations of features [14,15], novel phenotypes associated with genes known to cause other related genetic diseases [16,17], and other syndromic congenital disorders [18,19]. The absence of these disorders in other genetic databases like OMIM points to the rarity of these conditions. Our review also revealed 14 rare genetic disorders that have exclusively been described in the Lebanese population (Table 1). These include the Lebanese type of Mannose 6-Phosphate Receptor Recognition Defect (MIM ^#^ 154570), a form of autosomal recessive deafness (MIM ^#^ 603678), as well as a form of autosomal recessive dystonia (MIM ^#^ 612406). Two factors in combination could be responsible for the presence of this relatively large number of genetic conditions in this population. The first is the availability of and access to trained geneticists capable of recognizing and diagnosing these conditions, despite a relatively low grade medical and genetic infrastructure in the country [8]. The second factor is the persistent relatively increased prevalence of consanguineous marriages, resulting in the manifestation of rare disorders, many of which follow a recessive mode of inheritance [20]. In fact, around 65% of diseases we report in Lebanese subjects on CTGA follow an autosomal recessive inheritance pattern (Figure 2). This is in line with the numbers reported in 2015 [8]. Disorders following an autosomal dominant inheritance pattern make up about 26% of all reports, followed by X-linked and mitochondrial disorders.

We categorized the genetic disorders in the Lebanese population based on the WHO ICD-10 classification criteria (Figure 3). The most common category of disorders is congenital malformations, deformations and chromosomal abnormalities, followed by endocrine, nutritional and metabolic diseases and diseases of the nervous system. This pattern is comparable to data from other Arab countries in the CTGA database [21]. The overwhelming predominance of congenital malformations points to the large number of monogenic syndromic disorders in the database. Most of these disorders are relatively rare, with prevalence rates of less than 1 in 100,000. In fact, 314/394 (79.6%) Lebanese genetic disorders with known prevalence rates are rare, with rates less than 10 in 100,000. On the other hand, based on limited studies, certain disorders, including Fanconi anemia (MIM ^#^ 227650), alpha/beta thalassemia (MIM ^#^ 604131; MIM ^#^ 613985), and familial hypercholesterolemia (MIM ^#^ 143890), have been reported to have high prevalence rates [22,23,24,25]. Analyzing CTGA entries also allows us to note several rare disorders with a remarkably high occurrence among Lebanese subjects. For instance, odontoonychodermal dysplasia (MIM ^#^ 257980) and Dyggve-Melchior-Clausen syndrome (MIM ^#^ 223800) have each been identified in eight different families to date, while 15 unrelated families have been reported with Berardinelli-Seip congenital lipodystrophy type 2 (MIM ^#^ 269700). In its various forms, predominantly type 1A (MIM ^#^ 220290), recessive deafness has been reported in subjects from over 30 Lebanese families. The presence of such rare genetic disorders, especially in large consanguineous kindreds, is very useful in the mapping of the causative loci and the identification of the causal gene [26,27,28]. In fact, our survey on the Lebanese population identified 58 separate genetic disorders that were first mapped in Lebanese families (Table 1). Despite this, 62 of the disorders reported in Lebanon in the CTGA Database remain unmapped. 

We collated the clinical features of all Lebanese subjects in the CTGA Database (Table 2). Individual patients and familial studies frequently reported intellectual disability, developmental delay, short stature, hearing impairment, muscular hypotonia etc., highlighting the role of congenital malformations, in accordance with our ICD-10 classification. In contrast, an analysis of clinical manifestations in studies involving large groups of patients, mostly comprising association studies, brings out the impact of multifactorial and polygenic disorders that are common in the population (Table 2). For instance, diabetes, hypercholesterolemia, coronary artery disease, and neoplasms feature prominently.

The most recent review of genetic disorders on Lebanon which surveyed CTGA, OMIM, and the literature reported a total of 378 diseases reported in individuals of Lebanese origin [8]. In a preliminary 2017 CTGA analysis, only about half of these reported diseases had been molecularly diagnosed. In the current study, 78% of the 642 diseases we report have a molecular diagnosis. This rise in the number of reported diseases as well as in percentage of molecularly diagnosed cases is likely due to the increased adoption of NGS by Lebanese clinics/diagnostic centers. In addition, from our own experience, revisiting NGS data has helped in identifying causal variants of previously undiagnosed cases [29]. However, there remain 131 diseases out of the 642 total diseases we report here from CTGA wherein the Lebanese subjects lack a molecular diagnosis. The vast majority of these 131 disorders (82%) contain purely clinical descriptions or reports with no molecular study attempted.

To date, CTGA hosts a total of 1316 variant records described in Lebanese subjects. The majority of variants entered into CTGA expectedly involved substitutions, followed by deletions, duplications, as well as in-del and insertion variants. Less frequently reported variants included haplotypes, involving up to three variants on one allele, and microsatellites (Figure 4A). Variant records were added to CTGA and screened against dbSNP and ClinVar variant databases, revealing 150 CTGA-exclusive variants (Appendix A), 840 with both dbSNP and ClinVar records, 274 with dbSNP records, and 28 with ClinVar records (Figure 4B). Additionally, 222 variants were described in text summaries in disease and gene records, with over half of these involving copy number variations (54.8%), and the remaining entries reporting linkage studies, unspecified/whole gene deletions, gene rearrangements, inversions, as well as karyotypes. HLA and KIR alleles, as well as Gm and MHC class III allele variants were excluded from this count.

A more detailed look at the individual variants reported reveals several interesting results. The first of these is the presence of high prevalence variants reported in subjects of Lebanese origin. One such example is the p.Met1Ile mutation (rs587777839) in *PET100*. The latter has been identified in over 31 subjects exclusively from 12 different Lebanese families to date [30,31,32,33]. Another variant to note is the well-known p.Cys681X variant in the *LDLR* gene (rs121908031). Although it has been identified in various Arab and non-Arab nationalities, this variant, associated with Familial Hypercholesterolemia 1 (MIM ^#^ 143890), has been termed the Lebanese allele because of its high frequency in the Lebanese population [34,35] and even the Lebanese diaspora [36,37,38]. Evidence points towards founder mutation events driving the increased prevalence of these two variants.

Another category of variants are those that, although have been reported in non-Arab subjects, have so far only been reported in Lebanese subjects among Arabs in CTGA. For instance, a mutation in *SLC52A2* (rs398124641) leading to Brown-Vialetto-Van Laere Syndrome 2 (MIM ^#^ 614707) has been identified in two large unrelated Lebanese consanguineous families [39,40]. Another in *BSCL2* (rs587777608) has been reported in five unrelated Lebanese families with Congenital Generalized Lipodystrophy 2 (MIM ^#^ 269700) [41,42]. A p.Met1Val mutation in *DMP1* (rs104893834), associated with Autosomal Recessive Hypophosphatemic Rickets 1 (MIM ^#^ 241520) has been described in 14 subjects from at least three different Lebanese families [43,44,45].

Repositories of population specific genetic variants are crucial in providing clinical interpretations of these variants for both rare and common genetic disorders [46]. This is especially true for Arab nations which are burdened with a high incidence of rare genetic disorders and occurrence of founder mutations within their populations. Unfortunately, the Middle Eastern population is represented poorly in global variation databases, such as the Genome Aggregation Database [47]. Through continuous updates since its first release in 2005, CTGA is now the largest compendium of clinical genomic variants in Arab populations. Despite only hosting bibliographic data, 13% of the Lebanese variants within it are not found in either dbSNP or Clinvar, indicating its value for clinicians and researchers who deal with Arab patients (Appendix A).

CTGA is freely accessible online and researchers are encouraged to make use of the data available within it. As an example, here we show the type of data on ciliopathies in Lebanon that can be accessed from the database. CTGA contains information on 31 different ciliopathies that have been diagnosed in Lebanese subjects. These include several subtypes of Bardet-Beidl Syndrome, Leber Congenital Amaurosis and Usher Syndrome (Appendix A). Some of these ciliopathies are disorders that were first mapped in Lebanese families, such as Orofaciodigital Syndrome XIV (MIM ^#^ 615948), Short-Rib Thoracic Dysplasia 14 with Polydactyly (MIM ^#^ 616546), and Bardet-Biedl Syndrome 10, (MIM ^#^ 615987) [48,49,50]. On the other hand, the database also contains reports of rare ciliopathies in Lebanese subjects that are yet to be mapped, such as Ciliary Discoordination due to Random Ciliary Orientation (MIM ^#^ 215518) and Rhizomelic Dysplasia, Scoliosis, and Retinitis Pigmentosa (MIM ^#^ 610319) [51,52]. Of the 676 genes studied in Lebanese subjects, 28 (4.1%), carrying a total of 45 variants, are genes related to ciliopathies (Appendix A). Notably, seven of these variants, associated with subtypes of LCA, BBS, and Usher Syndrome, were absent from dbSNP and ClinVar.

## 4. Conclusions

In this report, we have attempted to provide an overview of the status of genetic disorders in Lebanon. High levels of consanguinity have been shown time and again to correlate with the spread of rare recessive disorders through inbred kindreds, especially in Arab populations [27,53,54,55]. In Lebanon, the relatively high level of consanguinity, coupled with a severe lack of genetic infrastructure within the country [8], has serious implications for the diagnosis and treatment of families affected with genetic disorders. There is thus an immediate need to both increase awareness among the general population on the consequences of inbreeding and familial genetic disorders, as well as to build advanced molecular diagnostic facilities. Simultaneously, despite improvement over the past 20 years, active effort needs to be made towards building an environment that further provides support for clinical research. 

Population level screening programs are still in their infancy in Lebanon. Although privately operated neonatal screening programs exist, they are estimated to cover less than half of the newborn population [56]. At the same time, rare disease registries that can offer valuable insights to clinicians, pharmaceutical companies and families of affected patients are non-existent. Efforts towards initiating a comprehensive public neonatal screening program and establishing rare disease registries could go a long way towards reducing the burden of genetic disorders in the country.

## Figures and Tables

**Figure 1 genes-12-01518-f001:**
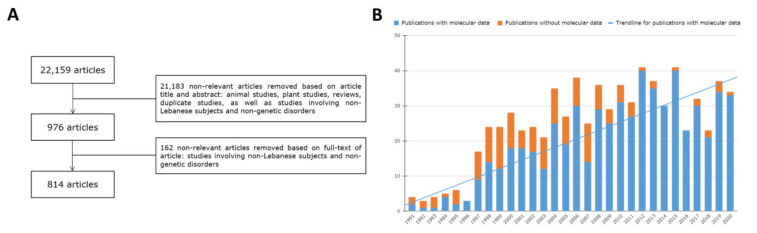
Bibliographic review of papers with Lebanese subjects. (**A**) Flowchart detailing criteria of study selection (**B**) Number of final selected publications (1991–2020) with and without molecular data classed by year of publication. Trendline is based on the number of molecular studies per year.

**Figure 2 genes-12-01518-f002:**
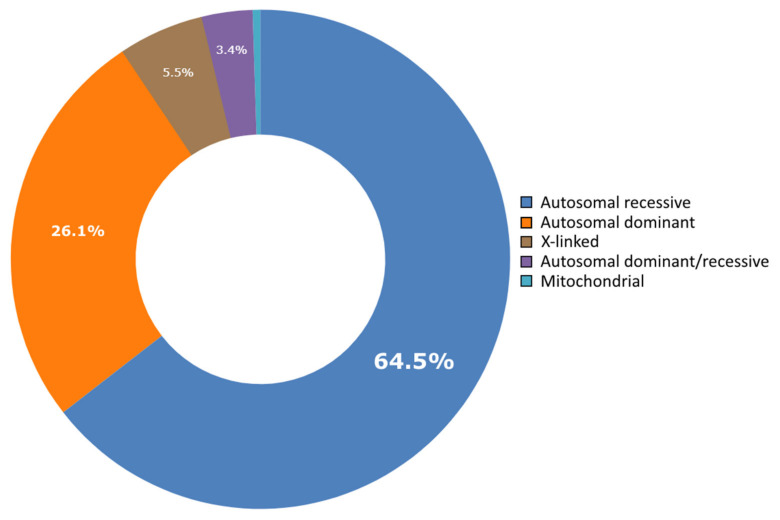
Distribution of mode of inheritance of diseases reported in Lebanese subjects in CTGA.

**Figure 3 genes-12-01518-f003:**
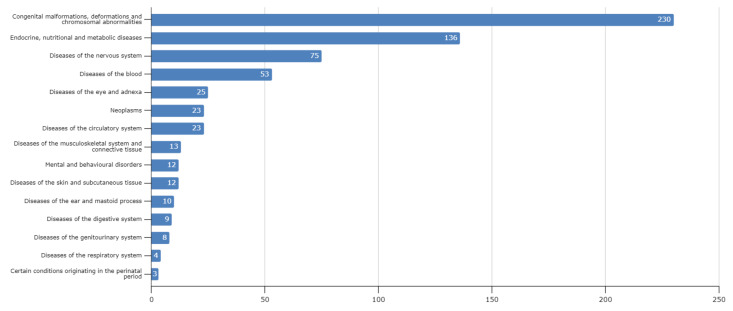
WHO ICD-10 classification of diseases reported in Lebanese subjects in CTGA. X-axis denotes number of disease entries in each class.

**Figure 4 genes-12-01518-f004:**
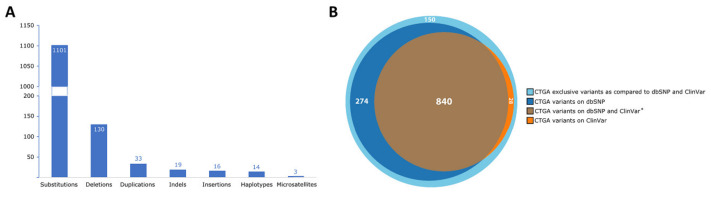
CTGA variant records reported in Lebanese subjects. (**A**) Distribution of types of 1316 CTGA variant records reported in Lebanese subjects. (**B**) Cylinder Venn distribution of CTGA variant records reported in Lebanese subjects relative to dbSNP and ClinVar variant databases (last updated 14 June 2021). * 24 of the 1316 variants were omitted due to disparate modes of classification between dbSNP and ClinVar, involving multiple distinct variants at the same position as well as haplotypes.

**Table 1 genes-12-01518-t001:** List of selected diseases in the CTGA Database with Lebanese subjects. Rows marked with * indicate diseases reported exclusively in Lebanese subjects. Rows marked with ^#^ indicate diseases that were first mapped in Lebanese subjects. Complete list of records is available in Appendix A.

Name	Phenotype OMIM Number	Related Gene Record	Gene/Locus OMIM Number
Diseases on CTGA not on OMIM (numbers in parentheses denote number of patients described)			
Brachytelephalangy with Mental Retardation, Peculiar Face and Short Stature (1 family, 2 patients)			
Congenital Contractures, Short Stature, Abnormal Face, Microcephaly, Scoliosis, Hip Dislocation, and Severe Psychomotor Retardation (2 families, 2 patients)			
Craniosynostosis, Telecanthus, Scalp Hair Abnormalities, and Sensorineural Deafness (1 family, 2 patients)			
Discoid Lupus Erythematosus (1 family, 4 patients)		*TRAF3IP3*	607043
Intellectual Deficiency, Unclassified (at least 147 patients)			
Linear and Whorled Nevoid Hypermelanosis with Cerebral Aneurysms (1 family, 1 patient)			
Marfanoid Habitus-Inguinal Hernia-Advanced Bone Age Syndrome (1 family, 2 patients)		*EFEMP1*	601548
Microcephaly, Colobomatous Micropthalmia, and mental Retardation (1 family, 2 patients)			
Multiple Anomalies, Mental Retardation, Megarbane-Le Merrer-El Kallab Type (1 family, 3 patients)			
Multiple Congenital Anomalies, Megarbane-Rassi Type (1 family, 1 patient)			
Multiple Congenital Anomalies, Mental Retardation, Ambiguous Genitalia, Microcephaly, Seizures, and Bone Malformations (1 family, 2 patients)			
Myeloproliferative Disorder, Unclassified (group of 69 patients)		*JAK2*	147796
Ptosis, Mental Retardation and 2/3 Toes Syndactyly (1 family, 2 patients)			
Pure Early-Onset Dementia Without Bone Cysts (1 family, 3 patients)		*TREM2*	605086
SOX11-Related Syndrome (1 family, 1 patient)		*SOX11*	600898
Tibial and Femoral Hypoplasia with ‘Hook’ Pelvis (1 family, 1 patient)			
TMTC3-Related Syndrome (1 family, 2 patients)		*TMTC3*	617218
12q24.31 Microdeletion Syndrome (1 family, 2 patients)			
8p23.1 Microdeletion Syndrome (1 family, 1 patient)			
Subacute Thyroiditis (1 family, 3 patients)		*HLA-B*	142830
Chromosome 10p Duplication Syndrome (1 family, 1 patient)			
Chromosome 7q Duplication Syndrome (1 family, 1 patient)			
Trisomy 17p (1 family, 1 patient)			
Chromosome 2p Duplication Syndrome (1 family, 1 patient)			
Diseases exclusively reported in Lebanon among Arab countries			
Acrodysostosis 2 With or without Hormone Resistance	614613	*PDE4D*	600129
Alveolar Soft Part Sarcoma	606243	*ASPSCR1; TFE3*	606236; 314310
Amyotrophy, Hereditary Neuralgic	162100	*SEPT9*	604061
Arrhythmogenic Right Ventricular Dysplasia, Familial, 11	610476	*DSC2*	125645
Arrhythmogenic Right ventricular Dysplasia, Familial, 9	609040	*PKP2*	602861
Atrial Septal Defect 4	611363	*TBX20*	606061
Atrial Septal Defect 5	612794	*ACTC1*	102540
Atrial Septal Defect, Secundum, with Various Cardiac and Noncardiac Defects *	603642		
Bardet-Biedl Syndrome 12	615989	*BBS12*	610683
Benign Chronic Pemphigus	169600	*ATP2C1*	604384
Bietti Crystalline Corneoretinal Dystrophy	210370	*CYP4V2*	608614
Borjeson-Forssman-Lehmann Syndrome	301900	*PHF6*	300414
Branchiogenic-Deafness Syndrome	609166		
Brown-Vialetto-Van Laere Syndrome 2 ^#^	614707	*SLC52A2*	607882
Brunner Syndrome	300615	*MAOA*	309850
Cardiomyopathy, Dilated, 1KK	615248	*MYPN*	608517
Cardiomyopathy, Dilated, with Hypergonadotropic Hypogonadism	212112		
Cataract 11, Multiple Types	610623	*PITX3*	602669
CDAGS Syndrome	603116		
Cerebral Creatine Deficiency Syndrome 1	300352	*SLC6A8*	300036
Char Syndrome	169100	*TFAP2B*	601601
Ciliary Discoordination due to Random Ciliary Orientation	215518		
Ciliary Dyskinesia, Primary, 3	608644	*DNAH5*	603335
Clouston Syndrome	129500		
Coffin-Siris Syndrome 4	614609	*SMARCA4*	603254
Combined Oxidative Phosphorylation Deficiency 1 ^#^	609060	*GFM1*	606639
Congenital Hemidysplasia with Ichthyosiform Erythroderma and Limb Defects	308050		
Cornelia de Lange Syndrome 5	300882	*HDAC8*	300269
Cutis Laxa, Autosomal Recessive, Type IA	219100	*ELN; FBLN5*	130160; 604580
Deafness, Autosomal Recessive 14 *^,#^	603678		
Developmental and Epileptic Encephalopathy 13	614558	*SCN8A*	600702
Developmental and Epileptic Encephalopathy 42	617106	*CACNA1A*	601011
Developmental And Epileptic Encephalopathy 63	617976	*CPLX1*	605032
Diamond-Blackfan Anemia 6	612561	*RPL5*	603634
Diastrophic Dysplasia	222600	*SLC26A2*	606718
Dislocated Elbows, Bowed Tibias, Scoliosis, Deafness, Cataract, Microcephaly, and Mental Retardation *	603133		
Dubowitz Syndrome	223370		
Dystonia 17, Torsion, Autosomal Recessive *^,#^	612406		
Dystonia, Childhood-Onset, with Optic Atrophy and Basal Ganglia Abnormalities	617282	*MECR*	608205
Ectodermal Dysplasia and Neurosensory Deafness	224800		
Ehlers-Danlos Syndrome, Arthrochalasia Type, 2	617821	*COL1A2*	120160
Ehlers-Danlos Syndrome, classic type, 1	130000	*COL1A1; COL5A1*	120150; 120215
Emery-Dreifuss Muscular Dystrophy 5, Autosomal Dominant	612999	*SYNE2*	608442
Enterocolitis	226150		
Epidermodysplasia Verruciformis, Susceptibility to, 2	618231	*TMC8*	605829
Epidermolysis Bullosa with Congenital Localized Absence of Skin and Deformity of Nails	132000	*COL7A1*	120120
Epilepsy, Nocturnal Frontal Lobe, 1	600513	*CHRNA4*	118504
Factor XI Deficiency	612416		
Familial Mediterranean Fever, Autosomal Dominant	134610	*MEFV*	608107
Fanconi Anemia, Complementation Group D1	605724	*BRCA2*	600185
Fanconi Anemia, Complementation Group E	600901	*FANCE*	613976
Fanconi Anemia, Complementation Group I	609053	*FANCI*	611360
Fanconi Anemia, Complementation group N	610832	*PALB2*	610355
Fever, Familial Lifelong Persistent *^,#^	228400		
Fibromatosis, Gingival, with Hypertrichosis and Mental Retardation	605400		
Frontotemporal Dysplasia and/or Amyotrophic Lateral Sclerosis 4	616439	*TBK1*	604834
Fructose Intolerance, Hereditary	229600	*ALDOB*	612724
Generalized Epilepsy with Febrile Seizures Plus, Type 7	613863	*SCN9A*	603415
Hymen, Imperforate	237100		
Hyperalphalipoproteinemia 1	143470	*CETP*	118470
Hypercarotenemia And Vitamin A Deficiency, Autosomal Recessive *	277350		
Hyperphenylalaninemia, BH4-Deficient, B	233910		
Hypophosphatemic Rickets, Autosomal Recessive, 1	241520	*DMP1*	600980
Ichthyosis, Congenital, Autosomal Recessive, 10	615024	*PNPLA1*	612121
Immunodeficiency 69 *^,#^	618963	*IFNG*	147570
Immunodeficiency with Defective T-Cell Response to Interleukin 1 *	243110		
Infantile Liver Failure Syndrome 2	616483	*NBAS*	608025
Inflammatory Bowel Disease 28, Autosomal Recessive ^#^	613148	*IL10RA*	146933
Inflammatory Skin and Bowel Disease, Neonatal, 1 *^,#^	614328	*ADAM17*	603639
Insulin-like Growth Factor I, Resistance to	270450	*IGF1R*	147370
Intellectual Development Disorder with Short Stature, Facial Anomalies and Speech Defects *^,#^	606220	*FBXL3*	605653
Intellectual Developmental Disorder 62	618793	*DLG4*	602887
Internal Carotid Artery, Spontaneous Dissection of	147820	*MTHFR*	607093
Joubert Syndrome 22	615665	*PDE6D*	602676
Koolen-De Vries Syndrome	610443	*KANSL1*	612452
Laurin-Sandrow Syndrome	135750		
Leber Congenital Amaurosis 7	613829	*CRX*	602225
Lentigines	150900		
Lethal Congenital Contracture Syndrome 7	616286	*CNTNAP1*	602346
Loeys-Dietz Syndrome 1	609192		
Loeys-Dietz Syndrome 5	615582	*TGFB3*	190230
Lujan-Fryns Syndrome	309520		
Lymphoproliferative Syndrome, X-Linked, 2	300635	*XIAP*	300079
Macrocephaly-Capillary Malformation	602501		
Mannose 6-Phosphate Receptor Recognition Defect, Lebanese Type *	154570		
Mental Retardation with Optic Atrophy, Facial Dysmorphism, Microcephaly, and Short Stature	609037		
Mental Retardation-Hypotonic Facies Syndrome, X-Linked, 1	309580	*ATRX*	300032
Mental Retardation, X-Linked, Syndromic, Christianson Type	300243	*SLC9A6*	300231
Metaphyseal Chondrodysplasia with Cone-Shaped Epiphyses, Normal Hair, and Normal Hands	609989		
Mitochondrial Complex I Deficiency, Nuclear Type 17 ^#^	618239	*NDUFAF6*	612392
Mitochondrial Complex III Deficiency, Nuclear Type 6 ^#^	615453	*CYC1*	123980
Mitochondrial Complex III Deficiency, Nuclear Type 7 ^#^	615824	*UQCC2*	614461
Mitochondrial Complex IV Deficiency, Nuclear Type 12 ^#^	619055	*PET100*	614770
Mitochondrial DNA Depletion Syndrome 11 ^#^	615084	*MGME1*	615076
Muscular Dystrophy-Dystroglycanopathy (Congenital With Impaired Intellectual Impairment), Type B, 1	613155	*POMT1*	607423
Muscular Dystrophy-Dystroglycanopathy (Congenital with Mental Retardation), type B, 6	608840	*LARGE1*	603590
Muscular Dystrophy, Limb-Girdle, Autosomal Recessive 10	608807	*TTN*	188840
Myoclonic Epilepsy, Congenital Deafness, Macular Dystrophy, and Psychiatric Disorders	604363		
Myofibrillar Myopathy 10 ^#^	619040	*SVIL*	604126
Myofibrillar Myopathy 11 ^#^	619178	*UNC45B*	611220
Myopathy, Lactic Acidosis, and Sideroblastic Anemia 2 ^#^	613561	*YARS2*	610957
Neurofaciodigitorenal Syndrome	256690		
Neutrophilic Dermatosis, Acute Febrile	608068	*MEFV*	608107
Night Blindness, Congenital Stationary, Type 1E ^#^	614565	*GPR179*	614515
Night Blindness, Congenital Stationary, Type 1H ^#^	617024	*GNB3*	139130
Noonan Syndrome 4	610733	*SOS1*	182530
Noonan Syndrome-Like Disorder with or without Juvenile Myelomonocytic Leukemia J	613563	*CBL*	165360
Occult Macular Dystrophy	613587	*RP1L1*	608581
Odontoonychodermal Dysplasia	257980	*WNT10A*	606268
Orofaciodigital Syndrome, Type IV	258860		
Osteogenesis Imperfecta, Type XVI ^#^	616229	*CREB3L1*	616215
Otopalatodigital Syndrome, Type I	311300	*FLNA*	300017
Paget Disease of bone 2, Early-onset	602080		
Pallister-Hall Syndrome	146510		
Parkinson Disease 7, Autosomal Recessive Early-Onset	606324		
Pentosuria	260800		
Peripheral Neuropathy, Autosomal Recessive, with or without Impaired Intellectual Development	618124	*MCM3AP*	603294
Pigmentary Disorder, Reticulate, with Systemic Manifestations	301220		
Pitt-Hopkins Syndrome	610954		
Premature Ovarian Failure 2B * ^#^	300604		
Progressive Familial Heart Block, Type IB ^#^	604559	*TRPM4*	606936
Pseudoachondroplasia	177170	*COMP*	600310
Ramon Syndrome	266270	*ELMO2*	606421
Retinopathy, Pigmentary, and Mental Retardation	268050	*VPS13A*	605978
Rhizomelic Dysplasia, Scoliosis, and Retinitis Pigmentosa *	610319		
Roifman Syndrome	616651	*RNU4ATAC*	601428
Short Stature and Facioauriculothoracic Malformations *	609654		
Short-Rib Thoracic Dysplasia 14 With Polydactyly ^#^	616546	*KIAA0586*	610178
Silver-Russell Syndrome	180860		
Skeletal Dysplasia, Rhizomelic, with Retinitis Pigmentosa *	609047		
Spasticity, Childhood-Onset, with Hyperglycinemia ^#^	616859	*GLRX5*	609588
Spinal Muscular Atrophy, Distal, Autosomal Recessive ^#^	607088	*VRK1*	602168
Spinocerebellar Ataxia 13	605259	*KCNC3*	176264
Spinocerebellar Ataxia 35	613908	*TGM6*	613900
Spinocerebellar Ataxia, Autosomal Recessive 2	213200	*PMPCA*	613036
Spinocerebellar Ataxia, Autosomal Recessive 24	617133	*UBA5*	610552
Spinocerebellar Degeneration and Corneal Dystrophy	271310		
Spondylocostal Dysostosis 3, Autosomal Recessive ^#^	609813	*LFNG*	602576
Chromosome 10q26 Deletion Syndrome	609813		
Spondylocostal Dysostosis, Autosomal Recessive 2 ^#^	608681	*MESP2*	605195
Spondyloepimetaphyseal Dysplasia, Maroteaux Type ^#^	184095	*TRPV4*	605427
Spondylometaphyseal Dysplasia, Megarbane-Dagher-Melki Type ^#^	613320	*PAM16*	614336
Stocco Dos Santos X-Linked Mental Retardation	300434	*SHROOM4*	300579
Surfactant Metabolism Dysfunction, Pulmonary, 1	265120	*SMDP1*	265120
Teeth, Supernumerary	187100		
Testes, Rudimentary	273150		
Thiopurines, Poor Metabolism of, 1	610460	*TPMT*	187680
Tricuspid atresia	605067	*NFATC1*	600489
Tubulointerstitial Kidney Disease, Autosomal Dominant, 2	174000	*MUC1*	158340
Ulnar Hypoplasia	191440		
Variegate Porphyria	176200	*PPOX*	600923
Vertebral, Cardiac, Renal, and Limb Defects Syndrome 2 ^#^	617661	*KYNU*	605197
Vibratory Urticaria	125630	*ADGRE2*	606100
Vitamin K-Dependent Clotting Factors, Combined Deficiency of, 2 ^#^	607473	*VKORC1*	608547
Diseases first linked or mapped in Lebanese subjects			
Alpha/Beta T-Cell Lymphoma with Gamma/Delta T-Cell Expansion, Severe Cytomegalovirus Infection, and Autoimmunity ^#^	609889	*RAG1*	179615
Bardet-Biedl Syndrome 10 ^#^	615987	*BBS10*	610148
Charcot-Marie-Tooth Disease, Demyelinating, Type 4F ^#^	614895	*PRX*	605725
Charcot-Marie-Tooth Disease, Type 4H ^#^	609311	*FGD4*	611104
Deafness, Autosomal Recessive 13 ^#^	603098		
Deafness, Autosomal Recessive 21 ^#^	603629	*TECTA*	602574
Deafness, Autosomal Recessive 9 ^#^	601071	*OTOF*	603681
Dihydropyrimidinase Deficiency ^#^	222748	*DPYS*	613326
Galloway-Mowat Syndrome 1 ^#^	251300	*WDR73; ZNF592*	616144; 613624
Hepatic Venoocclusive Disease with Immunodeficiency ^#^	235550	*SP110*	604457
Hydatidiform Mole, Recurrent, 1 ^#^	231090	*NLRP7*	609661
Hypercholesterolemia, Familial, 4 ^#^	603813	*LDLRAP1*	605747
Ichthyosis, Congenital, Autosomal Recessive 13 ^#^	617574		
Immunodeficiency 12 ^#^	615468		
Immunodeficiency 40 ^#^	616433	*DOCK2*	603122
Immunodeficiency 56 ^#^	615207	*IL21R*	605383
Lipodystrophy, Congenital Generalized, Type 2 ^#^	269700	*BSCL2*	606158
Pancreatic Agenesis 2 ^#^	615935	*PTF1A*	607194
Spinal Muscular Atrophy, Distal, Autosomal Recessive, 1 ^#^	604320	*IGHMBP2; REEP1*	600502; 609139
Weill-Marchesani Syndrome, Autosomal Recessive ^#^	277600	*ADAMTS10*	608990
Microphthalmia with Limb Anomalies ^#^	206920	*FNBP4; SMOC1*	615265; 608488
Geleophysic Dysplasia 1 ^#^	231050	*FBN1*	134797
Frank-Ter Haar Syndrome ^#^	249420	*SH3PXD2B*	613293
Orofaciodigital Syndrome XIV ^#^	615948	*C2CD3*	615944
Baller-Gerold Syndrome ^#^	218600	*RECQL4*	603780

* Diseases reported exclusively in Lebanese subjects. ^#^ Diseases that were first mapped in Lebanese subjects

**Table 2 genes-12-01518-t002:** Ten most prominent clinical features derived from HPO terms associated with Lebanese subjects in CTGA, ranked in decreasing order of occurrence. Ranking is based on the absolute number of individual subjects or subjects within a group, with each clinical feature.

Individual Subject Entries	Group Subject Entries
Intellectual disability	Diabetes
Hearing impairment	Coronary Artery Disease
Global developmental delay	Abnormal cholesterol levels
Seizures	Familial Mediterranean Fever
Short stature	Anemia
Muscular hypotonia	Inflammatory bowel disease
Delayed speech and language development	Hypertension
Failure to thrive	Obesity
Periodic fever	Myeloproliferative disorder
Atrial septal defect	Myocardial Infarction

## Data Availability

All data reported and analyzed in this study can be found at www.cags.org.ae/ctga-search.

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
