# Peer review of "Catalogue for Transmission Genetics in Arabs (CTGA) Database: Analysing Lebanese Data on Genetic Disorders"

_genes, 2021, doi:10.3390/genes12101518_

Round 1

Reviewer 1 Report

The manuscript regarding the CTGA database and common genetics disorders in individuals with Lebanese ethnicity is an interesting read that highlights the prevalence of genetic disorders in this specific population.  The paper is well reasoned and well constructed overall.

I have a few minor comments:

  1. Table 1: Please mark columns with phenotype MIM number and Gene/Locus MIM number respectively for easy understanding
  2. Table 1: Recommend referencing the disorders to appropriate publications
  3. A separate column with the number of individuals detected with each of those conditions will be helpful for the readers especially for those that are not on OMIM
  4. Figure 1: Recommend avoiding the terms X-linked dominant and recessive. Instead, it is better if you could combine them as X linked (See PMID: 15316978 for more info)
  5. I am curious to know the extent of genetic testing done for those disorders on CTGA but not on OMIM: Was exome/genome sequencing done or targeted panels?
  6. Line 223-224: Same comment as above. It would be helpful if you can provide information re: the extent of genetic testing done for these individuals with the undiagnosed 134 individuals.

Reviewer 2 Report

The manuscript from Bizzarri et al. describes an overview of the status of genetic  disorders in Lebanon based on the description of the Catalogue for Transmission Genetics in Arabs (CTGA) database (www.cags.org.ae/ctga). The population has an high incidence of birth defects, most genetics.

The database reports data on 641 genetic diseases and 676 15 related genes, described in Lebanese subjects. Authors described those exclusively described in the Lebanese population or reported in CTGA and not on OMIM.

All these data could be useful for necessary screening programs in Lebanon where high levels of consanguinity have been shown, correlating with the spread of rare recessive disorders.

The manuscript regards an argument interesting, albeit of limited relevance exclusively to the Lebanese region, in which surely a public neonatal screening program is necessary as well as a rare disease registry.

Some points for discussion are missing so that, at times, the manuscript looks like a simple description of the CTGA register without any critical review or useful considerations.

The discussioni is very unuseful and should be deepened considering aspects like high prevalence of specific variants, the presence of some variants only in Lebanese population and not in Arabian, the detection of private mutations, detected only in one family (private) and mostly the detection of disesases exclusively described in Lebanese people and absent in other countries (where the diagnosi has been done?)
